# Facial Fat Grafting (FFG): Worth the Risk? A Systematic Review of Complications and Critical Appraisal

**DOI:** 10.3390/jcm11164708

**Published:** 2022-08-11

**Authors:** Luigi Schiraldi, Gianluca Sapino, Joachim Meuli, Michele Maruccia, Mario Cherubino, Wassim Raffoul, Pietro G. di Summa

**Affiliations:** 1Department of Plastic, Reconstructive Surgery, University Hospital of Lausanne (CHUV), 46, 1011 Lausanne, Switzerland; 2Department of Plastic Reconstructive and Aesthetic Surgery, University of Bari Aldo Moro, 70121 Bari, Italy; 3Department of Biotechnology and Life Sciences, University of Insubria, 21100 Varese, Italy

**Keywords:** lipofilling, facial fat grafting, fat transfer

## Abstract

**Introduction:** Autologous fat is ideal soft tissue filler. It is easily accessible, biocompatible, cheap, and it provides both volume augmentation and skin quality improvement. Fat grafting has been used since 1893, but it has only gained widespread popularity since the development of modern liposuction by Colemann and Illouz in the 1980s. Every year more than half a million facial fat grafting procedures are carried out worldwide and the trend is rapidly increasing. Overall, general complications associated with facial fat grafting are assumed to be around 2%. Is that true? **Material and Methods**: Until July 2021, a systematic search of the literature was performed interrogating PubMed search engines. The following algorithm was used for the research: (fat graft OR lipofilling) AND face AND complications. Exclusion criteria applied hierarchically were review articles, not reporting recipient site complications; not in English and paediatric population. Abstracts were manually screened by LS, GS, JM and PDS separately and subsequently matched for accuracy. Pertinent full-text articles were retrieved and analysed and data were extracted from the database. The flow chart of article selection is described following the Preferred Reporting Items for Systematic Reviews and Meta-Analyses (PRISMA) statement. **Results**: In total, 462 papers were identified by PubMed search. A total of 359 were excluded: 38 papers were not in English, 41 were review articles, 279 articles did not report recipient site complications and 1 was not on human subjects. Average complication rate ranged from 1.5% to 81.4%. A total of 298 adverse events were identified: 40 (13.4%) intravascular injections, 13 (4.3%) asymmetry, 57 (19.1%) irregularities, 22 (7.4%) graft hypertrophy, 21 (7%) fat necrosis, 73 (24.5%) prolonged oedema, 1 (0.3%) infection, 6 (2%) prolonged erythema, 15 (5%) telangiectasia and 50 (16.8%) cases of acne activation. **Conclusions**: FFG related side effects could be resumed in three categories: severe, moderate, and minor. Severe (13.4%) side effects such as intravascular injection or migration require neurological or neurosurgical management and often lead to permanent disability or death. Moderate (38.3%) side effects such as fat hypertrophy, necrosis, cyst formation, irregularities and asymmetries require a retouch operation. Minor (48.3%) side effects such as prolonged oedema or erythema require no surgical management. Despite the fact that the overall general complication rate of facial fat grafting is assumed to be around 2%, the real complication rate of facial fat grafting is unknown due to a lack of reporting and the absence of consensus on side effect definition and identification. More RCTs are necessary to further determine the real complication rate of this procedure.

## 1. Introduction

Autologous fat is an ideal soft tissue filler. It is easily accessible, biocompatible and cheap, while providing both volume augmentation and skin quality improvement [1]. Such features made it a precious tool for both aesthetic and reconstructive purposes. The safety and efficacy of fat grafting has been largely studied in different body areas, especially the breast, where fat grafting has been extensively used for both augmentation and reconstructive procedures [2]. However, severe complications of fat grafting have been reported, especially in gluteal and facial areas [3,4], following intravascular injection or migration.

If in gluteal fat grafting the mechanism of intravascular injection is well understood and avoidance of intramuscular injection dramatically improves safety [5,6], no clear consensus exists concerning facial fat grafting (FFG), where the mechanism leading to vascular complications is partially unknown and no prevention and treatment [4] guidelines have been produced. According to I.S.A.P.S. statistics, more than half a million facial fat grafting procedures are performed worldwide every year and the trend is rapidly increasing, with a high satisfaction rate for both patient (91.1%) and surgeon (88.6%) [7].

Despite being a “hot topic”, the scientific literature is weak or biased in reporting complications, without clear risk quantification.

Indeed, since 1998, not a single case of intravascular injection has been reported among major studies on facial lipofilling [8], with almost all reports on vascular accidents being described in case reports or small case series [4]. On the other hand, minor complications such prolonged oedema or erythema are often considered to be normal and therefore underreported or not included in the complications list (while being potentially impactful for the patients, extending the social downtime and decreasing procedure related satisfaction).

If overall complications seem to reach 2% [8], this estimation misses a clear distinction on the kind of complications, their incidence and proportion depending on the anatomical zone treated or the technique used.

The aim of this systematic review is to critically analyse, define and classify FFG-related complications, to establish a clinical guidance for both the reconstructive and aesthetic surgeon, and to improve evidence-based patient care.

## 2. Materials and Methods

A systematic search of the literature was performed interrogating PubMed search engines from January 1965 until July 2021. The following algorithm was used for the research: *(fat graft OR lipofilling) AND face AND complications*. Exclusion criteria applied hierarchically were review articles, not reporting recipient site complications; not in English and paediatric population.

Abstracts were manually screened by LS, GS, JM and PDS separately and subsequently matched for accuracy. Pertinent full-text articles were retrieved and analysed and data were extracted on the database. The flow chart of article selection is described following the Preferred Reporting Items for Systematic Reviews and Meta-Analyses (PRISMA) statement.

We defined as minor complications all the side effects requiring light medical treatment or no treatment at all. When a complication required that a patient return to the operatory theatre it was defined as moderate. Finally, severe complications were all those leading to life endangerment or permanent disability.

## 3. Results

A total of 462 papers were identified by PubMed search. Of which, 359 were excluded: 38 papers were not in English, 41 were review articles, 279 articles were not reporting recipient site complications, 1 was not on human subjects. A total of 103 [9,10,11,12,13,14,15,16,17,18,19,20,21,22,23,24,25,26,27,28,29,30,31,32,33,34,35,36,37,38,39,40,41,42,43,44,45,46,47,48,49,50,51,52,53,54,55,56,57,58,59,60,61,62,63,64,65,66,67,68,69,70,71,72,73,74,75,76,77,78,79,80,81,82,83,84,85,86,87,88,89,90,91,92,93,94,95,96,97,98,99,100,101,102,103,104,105,106,107,108,109,110,111,112] papers were finally included for data extraction (Figure 1).

A total of 5479 patients were included in the quantitative analysis, 5016 (91.5%) were females and 463 (8.5%) were males. Mean age (±SD) was 39.18 (±5.8 years) (Table 1). In 37 papers fat grafting was done with reconstructive purposes, in 45 with aesthetic ones, in 21 for both purposes. Mean (±SD) injected fat was 29.01 (±5.56 mL) (Table 2).

The average global complication rate ranged from 0% to 81.4% depending on publications. A total of 354 (out 4579 patients) adverse events were identified: 87 (24.6%) intravascular injections, 73 (20.6%) prolonged oedema, 57 (16.1%) recipient site irregularities, 50 (14.1%) cases of acne activation, 30 (8.5%) fat necrosis or lipogranuloma, 22 (6.2%) graft hypertrophy, 15 (4.2%) telangiectasia, 13 (3.7%) asymmetry, 6 (1.7%) prolonged erythema and 1 (0.3%) infection (Table 3).

After a global analysis of complications, and acknowledging the lack of consensus on FFG complications, we could regroup complications in three categories: severe, moderate and minor.

### 3.1. Severe Complications/Intravascular Injection

These complications included intravascular injection or migration, required neurological or neurosurgical management. Our search found 87 described cases of severe complications and it represented the most reported complication with almost a third of cases. Only two cases (2.3%) showed full neurological recovery. The highest rate of severe (vascular) complications occurred in cases of multisite injections (16–18.4%) and Glabella treatment (16–18.4%), followed by the forehead (10–11.5%) and temporal area (8–9.2%), which carried a medium-high risk. The peri-ocular region, nose and naso-labial folds carried a medium-low risk (5–5.7%, 4–4.6% and 4–4.6% risk, respectively). The safest facial zone to inject was the cheek (1–1.1% risk) (Figure 2).

### 3.2. Moderate Complications/Graft Related Complications

We defined as moderate all complications requiring supplementary surgical management.

These included fat hypertrophy, fat necrosis, cyst formation, irregularities and asymmetries, and accounted for 34.8% of total complications. Due to heterogeneous reporting, no correlation analysis was possible between specific areas, amount of fat grafted and specific complications rate.

### 3.3. Minor Complications

We defined as minor complications those that were considered as unattended side effects (40.6%) such as prolonged oedema or erythema, telangiectasia and acne reactivation not requiring surgical management. In different studies, minor complications varied from 0% [9] to 81.4% [8,10].

## 4. Discussion

This work highlights how severe complications represent without any doubt the most dramatic potential consequences of FFG, clashing with the worldwide diffusion and relative simplicity of the procedure.

Intravascular injection, requiring neurological or neurosurgical management, often leads to permanent disability or eventually death [4]. The exact rate of intravascular injection or migration remains unclear. Our search found 87 described cases of severe complications and it represented the most reported complication with almost a third of cases.

The presence of fat tissue in the bloodstream is always pathological and it is important to understand the exact mechanism leading to fat embolization. According to Cardenas-Camarena et al. [113], the fat can enter into the blood stream in two different fashions during a fat grafting surgery. This difference in aetiopathogenesis generates two distinct pathologies: microscopic fat embolism (MIFE), commonly called fat embolism syndrome (FES), and macroscopic fat embolism (MAFE) [114]. When fat enters microscopically (MIFE), it produces the so-called FES, whereas when it enters macroscopically, it produces the direct occlusion of blood vessels and causes a MAFE [115], as shown by reports presenting this condition [113,114]. Due to small volumes involved in facial fat grafting, MAFE is clinically more relevant but MIFE cannot be excluded.

MAFE’s clinical manifestation mimics a standard thromboembolism secondary to a blood clot or cholesterol plaque. The recovery rate of major complication (MAFE) is really poor. Only two cases of cerebral and retinal [4] showed a full recovery after fat embolization.

In the first case [116], a 38 year-old patient underwent fat injection for soft tissue augmentation in the frontal and temporal area 2 years after trauma. The surgeon utilised a 20 G blunt cannula. At the end of the treatment, the patient has ocular pain and flashes. The operator immediately suspected a fat embolization and addressed him to the ophthalmology ward. Time to treatment was less than 20′. Ocular massage and medical treatment could restore vision in 90′.

In the second case [117], a 22 year-old woman received bilateral temporal augmentation with autologous fat (25 mL per side) and 4 h later presented to the emergency department with right major stroke. A cerebral CT-Scan revealed multiple fat embolisms in the right internal carotid artery and middle cerebral artery with basal ganglia ischemia. She underwent a percutaneous mechanical lipectomy of the middle cerebral artery. Two hours after the procedure, the patient’s symptoms improved and at a 3-month follow-up visit, she was asymptomatic.

Importantly, when analysing prospective studies and RCTs, episodes of intravascular injections were completely unreported, underlining a target/aim publication bias. Such studies focused on general outcomes including graft take and minor and moderate common complications, but did not mention major complications such as intravascular injection [118].

Articles that focused on major complications were case reports or case series on a limited number of patients, differing drastically from outcome-oriented publications (which focused on the beneficial effects of fat graft on large cohorts).

According to I.S.A.P.S data, a total of 2.932.618 FFG were performed worldwide from 2015 to 2019. In the same time frame, according to our work, a total of only 15 severe complications were reported. Following these data, intravascular injection or migration should be considered very rare and almost anecdotal (roughly one in 5 million). Still, it is impossible to know if all the cases of severe complications have been reported, and this incidence seems defective.

Moreover, this review underlines how reporting articles on FFG are particularly inhomogeneous and how consensus in defining FFG-related complications is lacking. After reviewing the articles, we regrouped complications into different categories, hopefully simplifying future data reporting on the subject.

On the other hand, after critical paper analysis [4,119,120], we were able to summarise best practice technical tips of effective and safe FFG, consistent with that previously reported by K. Wang et al. Indeed, after a deep anatomical understanding of face perfusion, prevention is crucial in increasing FFG safety.

Measures to lower the risk of fat embolization were integrated in our institutional guidelines, as reported below.

Always use a blunt needle (cannulas) with as large a diameter as possible (at least 18 G);Use small aliquots of epinephrine on the recipient site before harvesting the fat;Always inject retrograde;Always choose the safest plan (know the depth of big vessels, particularly in external/internal carotidal circulation) and, if possible, follow U.S. guidance;Avoid excessive single point injection dose (<0.1 mL);Use 1 mL syringes.

## 5. Conclusions

Although the overall general complication rate of facial fat grafting is assumed to be around 2% [8], the real complication rate of facial fat grafting is unknown due to a lack of reporting and the absence of consensus on side effect definition and identification. A potential tendency to avoid reporting complications, especially for potentially devastating vascular embolization, cannot be formally excluded. More prospective studies are necessary to further determine the real complication rate of this procedure, with clear reporting of complication guidelines.

This review underlined how weak reporting of minor complications was found in the literature, probably as these are often not reported as they are considered para-physiological events. In cases of prolonged oedema or erythema, based on our clinical experience and the main reviews [7,121], we suggest considering it to be normal for a period of 14 to 21 days.

## Figures and Tables

**Figure 1 jcm-11-04708-f001:**
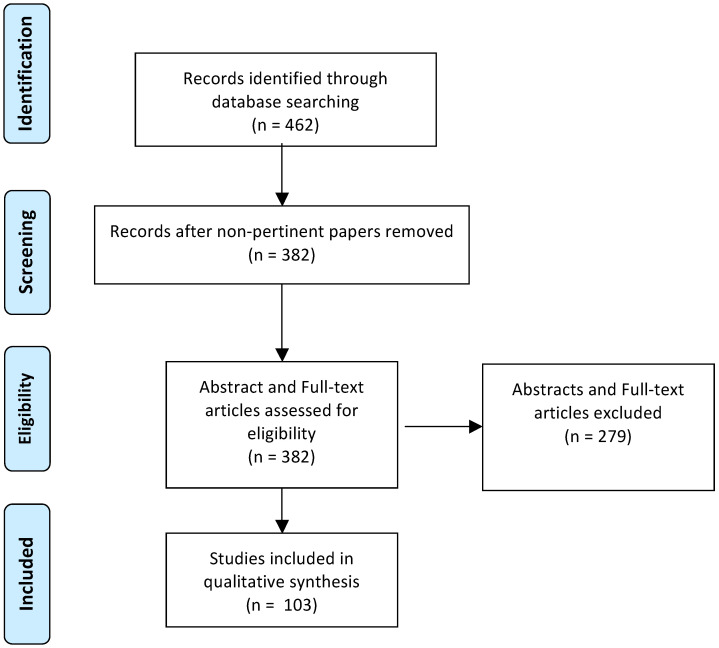
Flowchart according to the PRISMA criteria.

**Figure 2 jcm-11-04708-f002:**
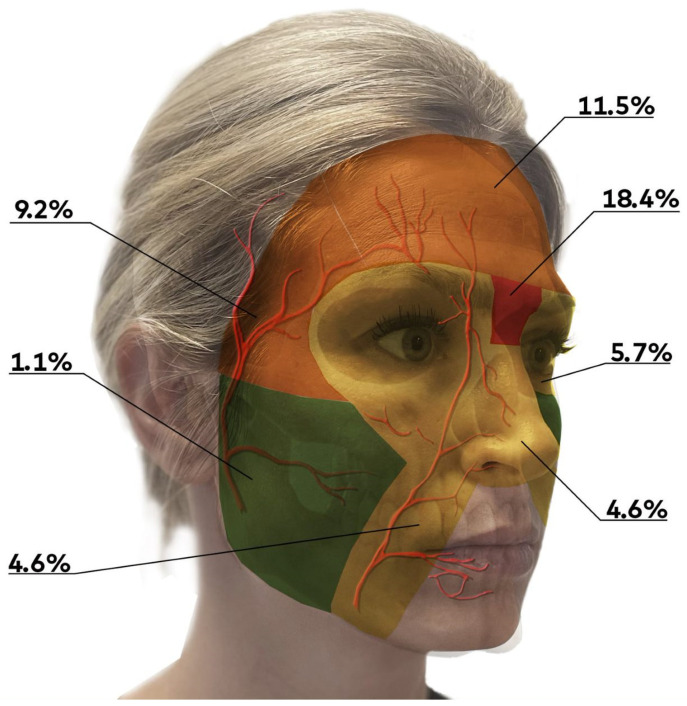
a Risk-based facial map.

**Table 1 jcm-11-04708-t001:** shows patients demograpic.

Demography	n 5479
** **♀** **/** **♂** **	5016/463
**Age (y)**	39.18 ± 5.8
**Graft (mL)**	29.01 ± 5.56

**Table 2 jcm-11-04708-t002:** shows article purposes.

Purpose	n 103
**Aesthetic**	45 (43.6%)
**Reconstruction**	37 (35.9%)
**Both**	21 (20.4%)

**Table 3 jcm-11-04708-t003:** resumes complications.

Complications	n 354	%
**Intravascular injections**	87	24.6
**Asymmetry**	13	3.7
**Irregularities**	57	16.1
**Graft hypertrophy**	22	6.2
**Fat necrosis or lipogranuloma**	30	8.5
**Infection**	1	0.3
**Prolonged oedema**	73	20.6
**Prolonged erythema**	6	1.7
**Telangiectasia**	15	4.2
**Acne activation**	50	14.1

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
