# Peer review of "Facial Fat Grafting (FFG): Worth the Risk? A Systematic Review of Complications and Critical Appraisal"

_jcm, 2022, doi:10.3390/jcm11164708_

Round 1
Reviewer 1 Report
Dear Authors
Your paper reads well. It is an addition to the 2019 paper in JPRAS by Gornitsky et al. I would remove the institution guidelines for fat grafting. It reflects an expert opinion. Instead, please add relevant guidelines or suggestions from other studies. In addition, a table with pertinent studies would be helpful.
Author Response
Thank you for your comment.
Very few paper in literature addressed clearly safety manoeuvres during fat grafting. In all these papers the authors gave their expert opinion, except one paper by Transatit et al. who investigated the mechanics of migration in a cadaveric study. In the annexed table the reviewer can find the papers that inspired our institutional guidelines.
author |
journal |
year |
Suggestions |
Wang X, et al |
J Craniofac Surg. |
2018 |
-Low Pressure injection -Don’t use sharp needle -pre-tunneling before fat injection - Aspiration before injection |
Liu H, et al |
Aesthetic Plast Surg. |
2019 |
--Low Pressure injection -Don’t use sharp needle - Aspiration before injection - Avoid traumatized tissues
|
Wang K. Et al |
Ann Plast Surg. |
2021 |
-Low Pressure injection -Don’t use sharp needle -pre-tunneling before fat injection - Aspiration before injection -Inject while mooving canula’s tip - Avoid traumatized tissues
|
Tansatit T |
Aesthetic Plast Surg. |
2015 |
- Add pressure on orbital rim to superior nasal border (Cadaveric study). |
Reviewer 2 Report
-methodology: including only Pubmed as serch seems narrow as search
- the result part is too short, especiallt the severe complications part; more details about complications and their consequences would be appreciated
- discussion: very few articles are cited, especially for the major complications and supposely most frequent complications, only (4) is cited, which seems not significant enough, line 144/45 for ex
what consequences did the complications have etc
Author Response
-methodology: including only Pubmed as seArch seems narrow as search
Indeed we agree on the limited une of Pubmed, still, after a deeper analysis in Embase and GoogleScholar we could not find other paper suitable for inclusion in our review
-the result part is too short, especially the severe complications part; more details about complications and their consequences would be appreciated
- discussion: very few articles are cited, especially for the major complications and supposely most frequent complications, only (4) is cited, which seems not significant enough, line 144/45 for ex what consequences did the complications have etc
A new paragraph with added references have been added in discussion section with more details on complications and their management, according to reviewers request.
Text now reads
“The presence of fat tissue in the bloodstream it is always pathological and it is important to understand the exact mechanism leading to fat embolization. According to Cardenas-Camarena et al(11), the fat can enter into the blood stream in two different fashions during a fat grafting surgery. This difference in etiopthogenesis generates two distinct pathologies: microscopic fat embolism (MIFE) commonly called fat embolism syndrome (FES), and macroscopic fat embolism (MAFE)(12). When fat enters microscopically (MIFE), it produces the so-called FES, whereas when it enters macroscopically, it produces the direct occlusion of blood vessels and causes a MAFE(13) as shown by reports presenting this condition(11,12). Due to small volumes involved in facial fat grafting MAFE is clinically more relevant but MIFE cannot be excluded.
MAFE’s clinic manifestation mimics a standard thromboembolism secondary to a blood clot or cholesterol plaque. Recovery rate of major complication (MAFE) is really poor. Only two cases of cerebral and retinal(4) showed a full recovery after fat embolization.
- In the first case (SzantyrA,OrskiM,MarchewkaI. Ocularcomplications following autologous fat injections into facial area: case report of a recovery from visual loss after ophthal- mic artery occlusion and a review of the literature. Aesthetic Plast Surg. 2017;41) a 38 years old patient underwent fat injection for soft tissue augmentation in frontal and temporal area 2 years after trauma. Surgeon utilised a 20G blunt cannula. At the end of the treatment the patient has ocular pain and flashes. The operator immediately suspected a fat embolization and addressed him to the ophthalmology ward. Time to treatment was less than 20’. Ocular massage and medical treatment could restore vision in 90’.
- In the second case (ZhouK,CaiC. Thesuccessful mechanical lipectomy treatment of cerebral fat embolism following autologous fat injection. Plast Reconstr Surg Glob Open. 2019) a 22 years old women received bilateral temporal augmentation with autologous fat (25ml per side) and 4 hours later presented to emergency department with the clinic of right major stroke. A cerebral CT-Scan revealed multiple fat embolisms in right internal carotid artery and middle cerebral artery with basal ganglia ischemia. She underwent a percutaneous mechanical lipectomy of middle cerebral artery. Two hours after the procedure patient improved her symptomps and at 3 month follow up visit she was asymptomatic." 

DIscussion section, page 6/14